# Thermoplastic Polyurethane Derived from CO_2_ for the Cathode Binder in Li-CO_2_ Battery

**DOI:** 10.3390/nano14151269

**Published:** 2024-07-29

**Authors:** Haobin Wu, Xin Huang, Min Xiao, Shuanjin Wang, Dongmei Han, Sheng Huang

**Affiliations:** 1The Key Laboratory of Low-Carbon Chemistry & Energy Conservation of Guangdong Province/State Key Laboratory of Optoelectronic Materials and Technologies, School of Materials Science and Engineering, Sun Yat-sen University, 135 Xingang West, Guangzhou 510275, China; 2School of Chemical Engineering and Technology, Sun Yat-sen University, Zhuhai 519082, China

**Keywords:** Li-CO_2_ batteries, CO_2_-based TPU, binders

## Abstract

High-energy-density Li-CO_2_ batteries are promising candidates for large-capacity energy storage systems. However, the development of Li-CO_2_ batteries has been hindered by low cycle life and high overpotential. In this study, we propose a CO_2_-based thermoplastic polyurethane (CO_2_-based TPU) with CO_2_ adsorption properties and excellent self-healing performance to replace traditional polyvinylidene fluoride (PVDF) as the cathode binder. The CO_2_-based TPU enhances the interfacial concentration of CO_2_ at the cathode/electrolyte interfaces, effectively increasing the discharge voltage and lowering the charge voltage of Li-CO_2_ batteries. Moreover, the CO_2_ fixed by urethane groups (-NH-COO-) in the CO_2_-based TPU are difficult to shuttle to and corrode the Li anode, minimizing CO_2_ side reactions with lithium metal and improving the cycling performance of Li-CO_2_ batteries. In this work, Li-CO_2_ batteries with CO_2_-based TPU as the multifunctional binders exhibit stable cycling performance for 52 cycles at a current density of 0.2 A g^−1^, with a distinctly lower polarization voltage than PVDF bound Li-CO_2_ batteries.

## 1. Introduction

The escalating emissions of CO_2_ due to the excessive utilization of fossil fuels underscore the pressing need for the capture and utilization of CO_2_ as a crucial aspect of sustainable development [1,2]. Li-CO_2_ batteries, an electrochemical device that converts the chemical energy from the reaction between lithium metal and CO_2_ into electrical energy, characterized by notable theoretical specific capacity (1876 Wh kg^−1^) [3] and elevated discharge voltage (2.8 V) [4], emerge as promising candidates for high-capacity battery applications. The operational principle of Li-CO_2_ batteries is as follows [5,6]:(1)$$4\mathrm{Li}+3{\mathrm{CO}}_{2}⇋2{\mathrm{Li}}_{2}{\mathrm{CO}}_{3}+C\left(E^{⊖}=2.8\;V\right)$$
(2)$$\phi_{+}=\phi_{+}^{⊖}+\frac{RT}{4F}\ln\left({\left[{\mathrm{CO}}_{2}\right]}^{3}{\left[{\mathrm{Li}}^{+}\right]}^{4}\right)$$
where $\phi_{+}$ represents the electric potential of the battery, *T* signifies temperature, R denotes the gas constant, F represents the Faraday constant, [CO_2_] indicates the concentration of dissolved carbon dioxide, and [Li^+^] represents the concentration of lithium ions. Specifically, the concentration should be the interfacial concentration at cathode/electrolyte interfaces.

The discharge product of Li-CO_2_ battery cathode is thermodynamically stable and poorly conductive Li_2_CO_3_ [7,8,9,10], leading to the challenges of the high charging voltage, low cycling life, and poor rate performance of Li-CO_2_ battery [11,12,13]. Consequently, the cathode’s performance assumes paramount importance in determining the overall efficacy of Li-CO_2_ batteries [14,15]. The polymer binder of the cathode plays a pivotal role in mitigating volume fluctuations of the cathode material during charge and discharge cycles, preventing material detachment and reducing internal resistance [16,17,18]. Conventional PVDF binder in batteries suffers from poor electrical conductivity and ionic conductivity [19,20,21], resulting in significant overpotentials in Li-CO_2_ batteries. Additionally, PVDF lacks catalytic activity for cathodic reactions, hindering the decomposition of Li_2_CO_3_ discharge products and compromising cycling reversibility [22]; moreover, PVDF lacks self-healing properties, and during the cycling process of Li-CO_2_, the cathode is prone to failure and powder detachment due to the volume expansion caused by the formation and decomposition of Li_2_CO_3_, hindering the improvement of Li-CO_2_ battery cycling performance [23,24]. Therefore, the development of binders with high conductivity conducive to enhanced cycling performance of Li-CO_2_ batteries, and possessing self-healing properties, assumes paramount importance for their advancement.

In this study, CO_2_-based thermoplastic polyurethane (TPU), synthesized from CO_2_, propylene oxide and isocyanate, etc. [25,26,27], was employed to replace conventional polyvinylidene fluoride (PVDF) as the binder of CO_2_ cathode. CO_2_-based TPU possesses the following advantages: (1) CO_2_-based TPU exhibits excellent CO_2_ adsorption characteristics, effectively enhancing the CO_2_ adsorption capacity of the cathode in Li-CO_2_ batteries and increasing the discharge voltage. (2) CO_2_-based TPU can increase the interfacial concentration of CO_2_ at cathode/electrolyte interfaces, ensuring uniform formation of Li_2_CO_3_ as a discharge product on the cathode surface during discharge; as a result, the particle sizes of Li_2_CO_3_ are much smaller, lowering the energy barrier of decomposition and reducing the charging voltage of Li-CO_2_ batteries. (3) Due to the excellent CO_2_ adsorption characteristics of CO_2_-based TPU with urethane groups (-NH-COO-), the dissolution of CO_2_ into the electrolyte and its shuttle to the anode during cycling are effectively reduced, protecting the lithium metal of anode and improving the cycling performance of Li-CO_2_ batteries. (4) CO_2_ based TPU exhibits excellent self-healing properties. During the cycling process of Li-CO_2_ batteries, the volume expansion caused by the formation and decomposition of Li_2_CO_3_ easily leads to cathode failure and powder detachment. CO_2_-based TPU binder possesses self-healing properties, effectively resisting binder damage and failure caused by volume expansion, thereby improving the cycling performance of Li-CO_2_ batteries. As a result, Li-CO_2_ batteries based on CO_2_-based TPU binder achieved a discharge voltage of 2.0 V at a current density of 200 mA g^−1^, a charging voltage of 3.5 V in the first cycle, and a cycling life of 52 cycles.

## 2. Materials and Methods

### 2.1. Materials

Propylene oxide (PO 99%) was purchased from Energy Chemical, Beijing, China. 1,4-butanediol (BDO, 99%) was purchased from Aladdin, Shanghai, China, and dehydrated with a 3 Å molecular sieve to ensure that the water content was less than 200 ppm. Polypropylene glycol (PPG, M_w_ ≈ 1000) and Methylene-bis(4-cyclohexylisocyanate) (HMDI, 99%) were purchased from Aladdin. Electrolyte 1 M LiTFSI in TEGDME (99.90%) was purchased from Duoduo chemistry, Suzhou, China. Poly(vinylidene fluoride) (PVDF, HSV-900) was purchased from Dongfu chemistry, Shanghai, China. Lithium metal (19 mm diameter, 600 μm thickness) was purchased from Canrud, Dongguan, China. N-methyl-2-pyrrolidone (NMP, 99.80%) was purchased from Macklin, Shanghai, China. Carbon nanotubes (CNTs) were purchased from Jicang nanotechnology, Nanjing, China.

### 2.2. Synthesis of CO_2_ Based TPU

CO_2_-based TPU was synthesized by a one-pot two-step method [28,29,30] (Figure 1). The oligocarbonate diols (PPCDLs) were synthesized by the direct copolymerization of propylene oxide (PO) and carbon dioxide (CO_2_) with 1,4-butanediol (BDO) as the proton exchange agent. The pre-polymer of PU with terminal isocyanate groups was prepared by reacting the prepared PPCDLs with excess hexamethylene diisocyanate (HMDI) and polypropylene glycol (PPG). The calculated amounts of polycarbonate diols were introduced into a three-necked flask equipped with a vacuum source, a thermometer, and a nitrogen inlet. After adding hexamethylene diisocyanate, the temperature was gradually increased to 80 °C. After 2 h of reaction, the pre-polymer with terminal -NCO groups was obtained. Then, the calculated amount of 1,4-butanediol (BDO) was slowly added to the system as a chain extender. Solvent N,N-dimethylformamide (DMF) and catalyst tin octoate were added sequentially. The reaction was maintained at 80 °C for another 2 h. After cooling to room temperature, the mixture was slowly poured into methanol, and the product was precipitated at 80 °C under vacuum and dried for 24 h.

### 2.3. Characterization

The successful synthesis of TPU was characterized by ^1^H nuclear magnetic resonance (^1^H-NMR, Bruker AVANCE 500, Billerica, MA, USA) and Fourier-transform infrared spectroscopy (FTIR, Thermo Fisher Nicolet 6700, Waltham, MA, USA, Frontier resolution: 0.15 cm^−1^). The CO_2_-based TPU was investigated using differential scanning calorimetry (DSC, DSC200PC, Netzsch, Selb, Germany) from −30 to +120 °C performed at a heating rate of 10 °C min^−1^, a cooling rate of 10 °C min^−1^, and a holding time of 3 min in a flow of N_2_. The mechanical properties of TPU samples were tested using an auto tensile tester (C610) at a tensile rate of 50 mm min^−1^, room temperature of 25 °C, and a relative humidity of 45%. The 180° peel test was conducted using an auto tensile tester (C610) at a tensile rate of 300 mm min^−1^, a room temperature of 25 °C, and a relative humidity of 60%. The surface morphologies of the cathode and Li metal samples were observed by scanning electron microscopy (SEM, TESCAN ClARA, 500–5000×, Brno, Czech Republic). The presence of Li_2_CO_3_ after charging and discharging was detected using an X-ray diffractometer (XRD, Malvern Panalytic, Empyrean, Malvern, UK, 5° min^−1^, Mo K_α_ = 0.71073 Å) and Raman spectroscopy (Renishaw inVia, Wotton-under-Edge, UK, 532 nm laser, resolution < 1 cm^−1^). The CO_2_ adsorption capacities of PVDF and TPU were tested using specific surface area and porosity analyzer (BET, Micromeritics ASAP 2460, Malvern, UK) at 0 °C. The characterization of surface of Li metal was performed using X-ray photoelectron spectroscopy (XPS, Thermo Fisher ESCALAB Xi+, Waltham, MA, USA).

### 2.4. Preparation of CO_2_ Cathode

Graphene (80%), CNT (10%), and PVDF(HSV-900+) or TPU binders (10%) were mixed and grinded in NMP to form a uniform slurry. The prepared slurry was scraped on carbon paper by a blade and dried in vacuum oven at 60 °C for 24 h. After drying, the material-loaded carbon paper was punched into 10 mm-diameter circular discs as working electrodes, with each containing 0.09~0.12 mg of active material (the mass of graphene and CNTs).

### 2.5. Fabrication of Li-CO_2_ Batteries

In this study, 2032-type coin cells with holes at the positive side were used. Polytetrafluoroethylene (PTFE) was applied on cathode to reduce solvent volatilization. The cell fabrication was conducted in an argon (Ar)-filled glovebox. A Li metal anode (15.8 mm in diameter) and glass fiber separator (Whatman, Maidstone, UK) were employed. A total of 1 M LiTFSI in TEGDME was used as the electrolyte.

### 2.6. Linear Sweep Voltammetry (LSV)

LSV was employed to test the electrochemical window of TPU binder. The scanning range was set at 0–5 V, with a scanning speed of 1 mV s^−1^, utilizing a CHI604D electrochemical workstation (CH Instruments, Shanghai, China).

### 2.7. Electrochemical Impedance Spectroscopy (EIS)

An EIS test was conducted on the Li-CO_2_ battery after 30 cycles. The test employed a 10 mV amplitude, with a frequency range of 0.1–10^6^ Hz, and was performed at a temperature of 28 °C using a CHI604D electrochemical workstation.

### 2.8. Battery Performance Test of Li-CO_2_ Batteries

The Li-CO_2_ battery was placed inside a test chamber in an Ar atmosphere. All Ar gas was evacuated using a mechanical pump, followed by immediate injection CO_2_ at 0.12 MPa. Constant current charge–discharge cycles were conducted under conditions of 0.12 MPa of CO_2_, a room temperature of 28 °C in a constant temperature incubator (Yuejin SPX-2500-II, Shanghai, China), a current density of 0.2 A g^−1^, and a cut-off capacity of 1000 mAh g^−1^ (20 μA for 0.1 mg of active material, for example).

### 2.9. Open Circuit Voltage Test

The Li-CO_2_ battery was placed inside a test chamber in an Ar atmosphere that was replaced with CO_2_ at 0.12 MPa. The Li-CO_2_ batteries were statistically placed under 0.12 MPa CO_2_ and 28 °C for 90 h, and the open circuit voltage was tested.

### 2.10. Deep Discharge Test

The Li-CO_2_ battery was placed inside a test chamber in Ar atmosphere and replaced by CO_2_ of 0.12 MPa. The battery was continuously discharged at 28 °C, 0.2 A g^−1^ until the discharge voltage reached 1.7 V.

## 3. Results and Discussion

The structural schematic of TPU and the corresponding ^1^H-NMR peaks with their respective chemical bonds are shown in Figure 2a. Successful synthesis of CO_2_-based TPU is validated through ^1^H-NMR (Figure 2b), ^13^C-NMR (Figure 2c), H-H COSY (Figure 2d), and FTIR (Figure 2e). A small characteristic signal appears at 6.95 ppm, attributed to the amino group in the amine-terminated polyester segment. The peaks at 1.24, 4.3, and 5.0 ppm correspond to the hydrogen atoms in the poly(propylene carbonate) diol (PPCDL) segment [28]. The peaks at 1.1 and 3.5 ppm are assigned to the hydrogen atoms in the poly(propylene glycol) (PPG) segment [31,32]. Additionally, the peak at 1.20 ppm is attributed to the hydrogen atoms in the hexamethylene diisocyanate (HMDI) segment [33,34]. Based on the integration of ^1^H-NMR, the molar ratios of PPCDL, PPG, BDO, and HMDI in TPU can be calculated to be 60.59%, 14.22%, 9.82%, and 15.36%, respectively. The peaks at 154.9, 75.4, 72.9, and 29.6 ppm correspond to the carbon atoms in the PPCDL segment. The peaks at 32.0, 25.7, and 25.0 ppm are assigned to the carbon atoms in the PPG segment. The peak at 73.3, 72.4, 69.0, 17.3, and 16.3 ppm is attributed to the carbon atoms in the HMDI segment. The test result of H-H COSY is shown in Figure 2d. J_67_ and J_78_ are the couple effect of the hydrogen of the PPCDL segment. J_13_ is the couple effect of the hydrogen of the PPG segment. J_910_ is the couple effect of the hydrogen of the BDO segment. The infrared absorption spectrum (FTIR) of TPU is depicted in Figure 2e. Strong absorption peaks appear at 1260 and 1750 cm^−1^, corresponding to the stretching vibrations of the O-C=O and C-O groups, respectively. Peaks at 2900 and 2875 cm^−1^ are attributed to the stretching vibrations of -CH_3_ and -CH_2-_ groups, respectively. Furthermore, peaks at 3340 and 1540 cm^−1^ are assigned to the stretching vibrations of the -NH- and C-NH bonds, respectively. To verify the thermal stability of CO_2_-based TPU, we conducted DSC testing on CO_2_-based TPU. The DSC curve of CO_2_-based TPU is shown in Figure 2f. The black curve is the differential curve of DSC curve, and the peak of it represents the temperature of glass transition temperature (T_g_). No crystallization or melting peaks are observed for CO_2_-based TPU in the temperature range of −30 to +120 °C, demonstrating its stability under working conditions of Li-CO_2_ batteries. The T_g_ of CO_2_-based TPU is measured at 23.5 °C, suggesting its elasticity and bonding capability under normal working conditions.

The tensile test results of CO_2_-based TPU are shown in Figure 3a. The tensile strength of CO_2_-based TPU reaches 19 MPa, and the elongation at break achieves 427%, indicating outstanding mechanical properties. The 180° peel strength tests of the cathode with PVDF and TPU as the binders are, respectively, shown in Figure 3b,c. The peel strength of the cathode with PVDF reached 7.3 N m^−1^, whereas that with CO_2_-based TPU was 4.7 N m^−1^. The bonding strength of CO_2_-based TPU was slightly weaker than that of PVDF. The adsorption capacity of PVDF and TPU powders for CO_2_ was tested by BET in a mixed ice-water bath at 0 °C [35] (Figure 3d). PVDF exhibits almost no CO_2_ adsorption capacity at all partial pressures, whereas CO_2_-based TPU displays significant adsorption capacity due to the presence of urethane groups (-NHCOO-). This uniform absorption and enrichment of CO_2_ at cathode/electrolyte interfaces contribute to an increase in the discharge voltage in Li-CO_2_ batteries. Moreover, the resultant discharge product Li_2_CO_3_ exhibits smaller particle sizes, thereby reducing the energy barrier of decomposition and enabling lower charge voltages. Consequently, the electrolytes remain more stable at lower voltages, thereby enhancing the cycle life of Li-CO_2_ batteries. Subsequently, linear sweep voltammetry (LSV) is employed to characterize the electrochemical stability of CO_2_-based TPU (Figure 3e). Within the range of 0–5 V, the current remained within 0.3 μA, with no significant increase in current, indicating that TPU did not undergo electrochemical decomposition. Therefore, it can be concluded that TPU is electrochemically stable within the working voltage range of Li-CO_2_ batteries (1.8–4.4 V) without decomposition.

The CO_2_-based TPU exhibited excellent self-healing properties. Following incisions cut by a blade, the CO_2_-based TPU samples before and after self-healing at 60 °C were observed under an optical microscope at a magnification of 10× [36]. The scratches were significantly restored after 60 min of self-healing (Figure 4a–c). After 10 min of self-healing at 60 °C, the scratches show obvious recovery; after 60 min of healing, the scratches are essentially completely gone. The tensile strength of the TPU before cutting and after healing was measured, indicating a healing efficiency of 72.8% at 32 °C (Figure 4d). The self-healing mechanism of TPU involves the mobility of its chain segments. As temperature increases, the mobility of chain segments is enhanced, facilitating the reestablishment of hydrogen bonds between molecules. The outstanding self-healing performance of TPU can effectively mitigate powder shedding attributed to cathode volume variation.

We assembled Li-CO_2_ batteries utilizing PVDF and CO_2_-based TPU as binders, respectively, and conducted charge–discharge cycles at a constant current of 0.2 A g^−1^ and a cut-off capacity of 1000 mAh g^−1^ in CO_2_ atmosphere. The charge–discharge curves for the 1st cycle, 5th cycle, and final cycle are shown in Figure 5a,b. From the charging platform, the charging voltage of PVDF battery is around 3.8 V, while that of TPU battery is around 3.5 V, indicating that TPU binders reduces the charging voltage of Li-CO_2_ batteries. The discharge voltage of the first cycle of the PVDF battery is around 1.8 V, while that of the CO_2_-based TPU battery is around 2.0 V. This discrepancy arises due to the CO_2_ adsorption capacity of CO_2_-based TPU, which elevates the partial pressure of CO_2_ on the surface of the electrode’s active material; according to the Nernst equation (Equation (2)), the absorption characteristics of CO_2_-based TPU for CO_2_ enhance the discharge voltage of Li-CO_2_ batteries. Meanwhile, from Figure 5a,b, it can be seen that the cycle life of Li-CO_2_ batteries with CO_2_-based TPU as the binder is significantly higher than that of Li-CO_2_ batteries with PVDF. The cycle life of Li-CO_2_ batteries with PVDF and CO_2_-based TPU binder at 0.2 A g^−1^ at a cut-off capacity of 1000 mAh g^−1^ is shown in Figure 5c. The lifespan of Li-CO_2_ batteries with PVDF is only 400 h, while the lifespan for batteries with CO_2_-based TPU reaches 550 h, demonstrating the significant role of CO_2_-based TPU in extending the battery cycle stability of Li-CO_2_ batteries.

Electrochemical impedance spectroscopy (EIS) was conducted on Li-CO_2_ batteries with different cycles, as shown in Figure 6a. After the first cycle, both the EIS spectra of the Li-CO_2_ batteries with PVDF and TPU binders exhibited two semicircles, indicating that the SEI had not yet stabilized on either the cathode or the anode interfaces. After the fifth cycle, only one semicircle appeared, indicating that a stable SEI was formed on both the cathode and anode interfaces, resulting in similar interfacial characteristics. The Li-CO_2_ battery with PVDF as the binder exhibited a higher initial impedance of charge transfer resistance (R_ct1_) of 231.3 Ω and R_ct2_ = 386.4 Ω. In contrast, the Li-CO_2_ battery with CO_2_-based TPU as the binder showed a lower initial impedance of only R_ct1_ = 158.1 Ω and R_ct2_ = 362.2 Ω. The lower initial impedance of the TPU binder battery compared to the PVDF binder battery indicates that the Li-ion transfer impedance is smaller with CO_2_-based TPU as the binder. During the 5th and 10th cycles, the R_ct_ of the Li-CO_2_ batteries with PVDF binder increased to 638.3 Ω and 674.8 Ω, respectively, while the R_ct_ of the Li-CO_2_ batteries with TPU binder was only 504.2 Ω and 561.9 Ω, respectively. The Li-CO_2_ batteries with PVDF binder exhibited higher solution resistance (R_s_) values at the 1st, 5th, and 10th cycles of 26.9 Ω, 65.8 Ω, and 98.9 Ω, respectively. In contrast, the Li-CO_2_ batteries with TPU binder showed lower R_s_ values of 9.9 Ω, 57.81 Ω, and 45.2 Ω, respectively, suggesting that TPU plays a role in reducing the electrolyte consumption during SEI formation. The Li-CO_2_ batteries with PVDF binder exhibited a Warburg impedance of 660 Ω, 640 Ω, and 653 Ω at the 1st, 5th, and 10th cycles, respectively. In contrast, the Li-CO_2_ batteries with TPU binder showed a Warburg impedance of 652 Ω, 642 Ω, and 649 Ω at the same cycles. The nearly constant diffusion impedance indicates that TPU does not affect the lithium-ion conduction capability of the electrolyte. After 10 cycles, the Li-CO_2_ battery with CO_2_-based TPU binder exhibited significantly lower electrochemical impedance than the PVDF binder battery, suggesting the formation of a better solid electrolyte interface (SEI) with CO_2_-based TPU as the binder, which reduces the corrosion of the anode, thus preventing a sharp increase in impedance. We conducted rate performance tests on Li-CO_2_ batteries with PVDF and CO_2_-based TPU binders at a cut-off capacity of 1000 mAh g^−1^. The batteries underwent five cycles each at 0.2, 0.5, 1, and 2 C, with the rate performance of the Li-CO_2_ batteries assessed based on the overpotential (Figure 6b). The overpotential of the Li-CO_2_ batteries with CO_2_-based TPU was significantly smaller than those with PVDF, attributed to the CO_2_ adsorption capability of TPU, which increases the CO_2_ partial pressure on the surface of cathode, resulting in the higher discharge plateau of the battery. Additionally, the smaller Li_2_CO_3_ grains generated under higher CO_2_ partial pressure facilitate easier decomposition during charging. As a result, the Li_2_CO_3_ battery with a CO_2_-based TPU binder displayed a lower charging plateau, leading to reduced overpotential and enhanced rate performance.

Li-CO_2_ batteries with PVDF and CO_2_-based TPU binders were subjected to 0.12 MPa of CO_2,_ respectively, for 90 h, with the open circuit voltage measured under a CO_2_ atmosphere (Figure 6c). Under a 0.12 MPa CO_2_ atmosphere, the open circuit voltage of the Li-CO_2_ batteries with PVDF binder reached a maximum of 2.6 V, stabilizing at around 2.5 V whereas the Li-CO_2_ battery with the CO_2_-based TPU binder reached a maximum of 2.77 V, stabilizing at around 2.7 V. The higher open circuit voltage of the TPU bound Li-CO_2_ battery in the static state is attributed to the higher CO_2_ affinity of CO_2_-based TPU, resulting in a higher CO_2_ concentration on the surface of cathode, leading to a higher discharge plateau of the battery. Therefore, the open circuit voltage of the TPU binder Li-CO_2_ battery exceeded those of PVDF. The adsorption capacity of TPU for CO_2_ is stronger than that of PVDF, resulting in higher surface CO_2_ concentration on the cathode, which has a significant impact on the morphology of discharge products. Deep discharge tests were conducted on Li-CO_2_ batteries with PVDF and CO_2_-based TPU binders at a current density of 0.2 A g^−1^ (Figure 6d). The discharge plateau of the CO_2_-based TPU Li-CO_2_ battery reached 2.0 V, surpassing the 1.8 V of the PVDF battery, while the discharge capacity of TPU binder Li-CO_2_ batteries reached 8654 mAh g^−1^, higher than that of the PVDF battery (6173 mAh g^−1^). These results underscore the fact that the CO_2_ adsorption capacity of TPU increases the reaction concentration of CO_2_ on the cathode.

In order to investigate the influence of CO_2_-based TPU binder on the discharge products, we disassembled cycled Li-CO_2_ batteries after 30 cycles at 0.2 A g^−1^ and analyzed the cathode of PVDF and CO_2_-based TPU using an X-ray diffractometer (XRD) after sealing with vacuum mud. The XRD patterns of discharged PVDF and TPU binder Li-CO_2_ batteries are shown in Figure 7a, which exhibit obvious Li_2_CO_3_ peaks [37,38], which are the discharge product of Li-CO_2_ batteries. However, charged PVDF binder Li-CO_2_ batteries still exhibited distinct Li_2_CO_3_ peaks with lower intensities compared to the discharged state, indicating the insufficient reversibility of Li_2_CO_3_ at the PVDF binder CO_2_ cathode; During charging, incomplete decomposition of Li_2_CO_3_ into CO_2_ occurs, resulting in continuous accumulation of Li_2_CO_3_ at the cathode during cycling. This accumulation compromises the active material and ultimately leads to Li-CO_2_ battery failure. In contrast, the XRD patterns of CO_2_-based TPU binder Li-CO_2_ batteries show no peaks of Li_2_CO_3_ post-charging, signifying complete decomposition of Li_2_CO_3_ into CO_2_ in the presence of TPU binder. Therefore, Li_2_CO_3_ does not accumulate on cathode active sites, prolonging the cycle life for TPU binder Li-CO_2_ batteries, in line with the results in Figure 5. Similarly, the charged and discharged PVDF and CO_2_-based TPU binder Li-CO_2_ batteries were subjected to Raman tests under air-isolated conditions. The Raman test results of Li metal are shown in Figure 7b and are consistent with the results of X-ray diffraction. PVDF binder Li-CO_2_ batteries retained distinct Li_2_CO_3_ Raman peaks post-charging, whereas the Li_2_CO_3_ Raman peaks [39,40] vanished completely post-charging in TPU binder Li-CO_2_ batteries, underscoring the role of TPU in promoting the reversible decomposition of Li_2_CO_3_ and facilitating the reversible cycling of Li-CO_2_ batteries. Subsequently, the cathode sheets of TPU and PVDF binder Li-CO_2_ batteries after 30 cycles at 0.2 A g^−1^ were characterized via SEM. In the discharged state, the surfaces of PVDF (Figure 7e) and TPU (Figure 7c) cathode sheets exhibit notable differences. Li_2_CO_3_ clusters on the PVDF sheet are large and clustered, with clusters reaching 10 μm in diameter, whereas Li_2_CO_3_ on the TPU sheet is uniformly distributed on carbon fibers without clustering. In the charged state, Li_2_CO_3_ residues persist on the PVDF sheet (Figure 7f) albeit with smaller clusters compared to the discharged state, approximately 8 μm in size, whereas Li_2_CO_3_ is nearly absent on the CO_2_-based TPU sheet (Figure 7d) in the charged state. This further underscore CO_2_-based TPU’s efficacy in promoting reversible Li_2_CO_3_ decomposition, effectively mitigating the loss of active sites due to Li_2_CO_3_ deposition on the cathode, confirming its pivotal role in Li-CO_2_ battery cycling stability. This observation aligns with the TPU’s superior cycling performance compared to PVDF, as shown in Figure 5. The SEM images confirm that Li_2_CO_3_ in PVDF binder cathode cannot be completely reversibly decomposed during cycling, while Li_2_CO_3_ in TPU binder cathode sheets can maintain good reversibility during cycling.

To assess the impact of CO_2_-based TPU binder on the anode corrosion during cycling, PVDF and CO_2_-based TPU binder-based Li-CO_2_ batteries were disassembled after 30 cycles in an Ar atmosphere glove box, and the composition of the anode surface was characterized by XPS. From the XPS carbon spectra, Li_2_CO_3_ peaks [41,42] were observed on the surface of the anode in the PVDF binder-based Li-CO_2_ battery (Figure 8d), indicating anode corrosion. In contrast, no Li_2_CO_3_ peak was observed on the surface of the anode in the TPU binder-based Li-CO_2_ battery (Figure 8a), indicating that the anode remained intact without corrosion. Meanwhile, peaks corresponding to organic nitrides and organic fluorides were observed on the TPU binder-based battery, indicating the presence of organic fluoride and nitride compounds in the SEI layer. These components play an important role in protecting the lithium metal from corrosion, enhancing the cycling stability and lifespan of Li-CO_2_ batteries. Furthermore, from the XPS oxygen spectra, it is evident that the Li_2_CO_3_ signal peaks appeared prominently on the surface of the anode of PVDF binder-based Li-CO_2_ battery (Figure 8e), while no Li_2_CO_3_ signal peak was observed on the surface of anode of the TPU binder Li-CO_2_ battery (Figure 8b), further confirming the ability of CO_2_-based TPU binder to protect lithium metal from corrosion. This is attributed to the strong affinity of TPU towards CO_2,_ which effectively reduces the dissolution of CO_2_ into the electrolyte, preventing CO_2_ diffusion to the lithium metal during cycling, thereby reducing corrosion of CO_2_ and lithium metal at the anode surface, ultimately achieving the goal of protecting the lithium metal and enhancing the cycling stability and lifespan of Li-CO_2_ batteries. Moreover, from the XPS fluorine spectra, it can be observed that both LiF [43] and LiTFSI [44] are the main forms of fluorine present at the surface of anode of PVDF (Figure 8f) and TPU (Figure 8c) binder-based Li-CO_2_ batteries. However, the ratio of LiF peak area to LiTFSI peak area is higher for TPU binder-based batteries, indicating that a higher content of LiF protects the lithium metal in the SEI layer of TPU binder-based batteries. As a result, TPU binder-based Li-CO_2_ batteries have better cycling stability than PVDF binder-based batteries.

To visually assess the differences in anode morphology of Li-CO_2_ batteries with PVDF and TPU binders after 30 cycles at 0.2 A g^−1^, the cycled Li-CO_2_ batteries were disassembled inside a glovebox and the anode surfaces were examined via SEM. The surface of the lithium metal in the TPU binder batteries (Figure 9c,d) displayed a smooth and uniform morphology without noticeable defects. In contrast, the surface of the lithium metal in the PVDF binder batteries (Figure 9a,b) exhibited severe corrosion and the formation of prominent dendrites. This observation aligns with the conclusions drawn from XPS spectra, suggesting that the CO_2_-based TPU binder promotes the formation of an SEI passivation layer rich in LiF, which effectively shields the lithium metal from corrosion by CO_2_ during cycling. Moreover, the SEI facilitates a more uniform distribution of lithium ions reduced to lithium metal on the lithium metal surface, effectively reducing dendrite growth. This significantly contributes to the lower impedance observed in TPU binder Li-CO_2_ batteries compared to PVDF binder Li-CO_2_ batteries. CO_2_-based TPU hinders the dissolution of CO_2_ into the electrolyte, preventing corrosion of the lithium metal, and facilitates the formation of a more effective SEI protective layer. These combined effects enhance the cycling stability and lifespan of Li-CO_2_ batteries. This also explains the longer cycle life observed in TPU binder Li-CO_2_ batteries (Figure 5), as well as the lower impedance observed in TPU binder Li-CO_2_ batteries (Figure 6a), with impedance growth during cycling that is significantly lower than that of PVDF binder batteries.

## 4. Conclusions

In summary, by replacing the commonly used commercial polyvinylidene fluoride (PVDF) binder with a CO_2_-based thermoplastic polyurethane (TPU) binder, the stable cycling life of Li-CO_2_ batteries has been increased from 40 cycles with PVDF binder to 52 cycles with TPU binder, which reach a high level for catalyst-free Li-CO_2_ batteries [45,46,47]. This enhancement can be attributed to several key factors: (1) TPU, characterized by -NH- bonds, exhibits Lewis basicity, possessing stronger CO_2_ adsorption capability. This enhances the adsorption of CO_2_ at the cathode, reducing the diffusion of CO_2_ to the anode and its side reactions with lithium metal. (2) The CO_2_-based TPU binder enhances the cycling stability and lifespan of Li-CO_2_ batteries. TPU reinforces the CO_2_ adsorption capacity of the cathode, elevating the discharge voltage plateau of Li-CO_2_ batteries. Additionally, the generated Li_2_CO_3_ particles are smaller, facilitating more reversible cycling during charging and reducing the charging plateau of Li-CO_2_ batteries. (3) TPU promotes the formation of a dense SEI rich in LiF on the anode. This SEI layer enables uniform deposition and stripping of lithium ions at the anode during cycling, effectively reducing dendrite formation. Moreover, the dense SEI layer prevents direct contact between dissolved CO_2_ in the electrolyte and the lithium metal, thereby reducing the side reactions between CO_2_ and lithium metal, thus extending the cycling stability and life of Li-CO_2_ batteries.

## Figures and Tables

**Figure 1 nanomaterials-14-01269-f001:**
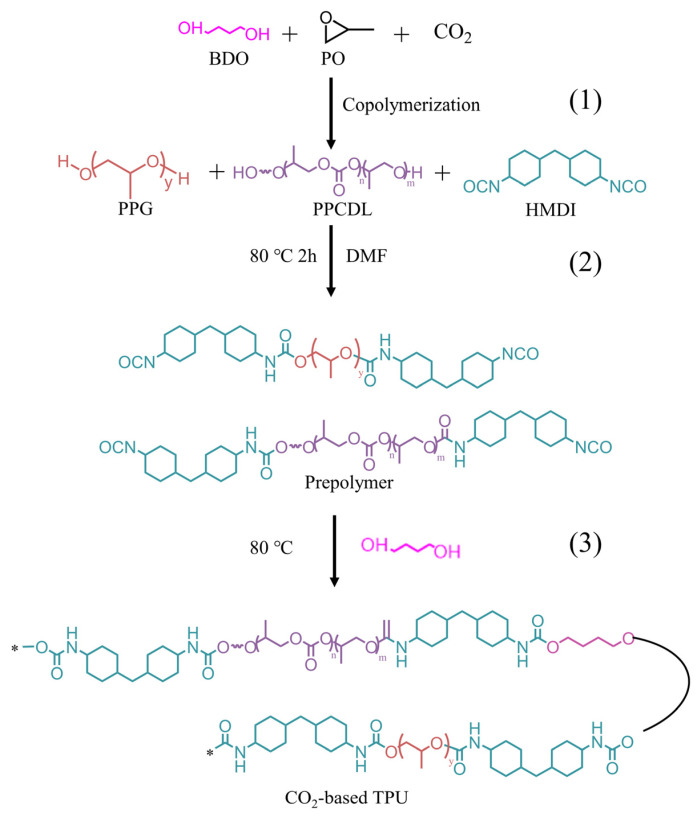
Schematic diagram of synthesis of CO_2_-based TPU using CO_2_, PO, BDO, PPG, and HMDI as raw materials. (**1**) Synthesizing PPCDL by copolymerization of BDO, PO, and CO_2_; (**2**) preparing the PU prepolymer by reacting PPG, PPCDL, and HMDI at 80 °C; (**3**) employing BDO as a chain extender to synthesize TPU at 80 °C.

**Figure 2 nanomaterials-14-01269-f002:**
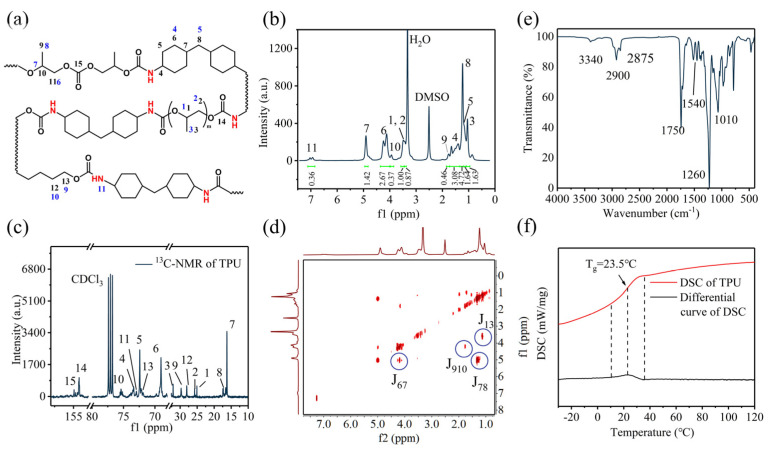
(**a**) Structure of TPU with labels for ^1^H-NMR (blue) and ^13^C-NMR (black); (**b**) ^1^H-NMR spectrum of TPU with DMSO-d6; (**c**) ^13^C-NMR spectrum of TPU with CDCl_3_; (**d**) H-H COSY spectrum of TPU; (**e**) FTIR spectrum of TPU; and (**f**) DSC curves of TPU recorded from −30 to +120 °C.

**Figure 3 nanomaterials-14-01269-f003:**
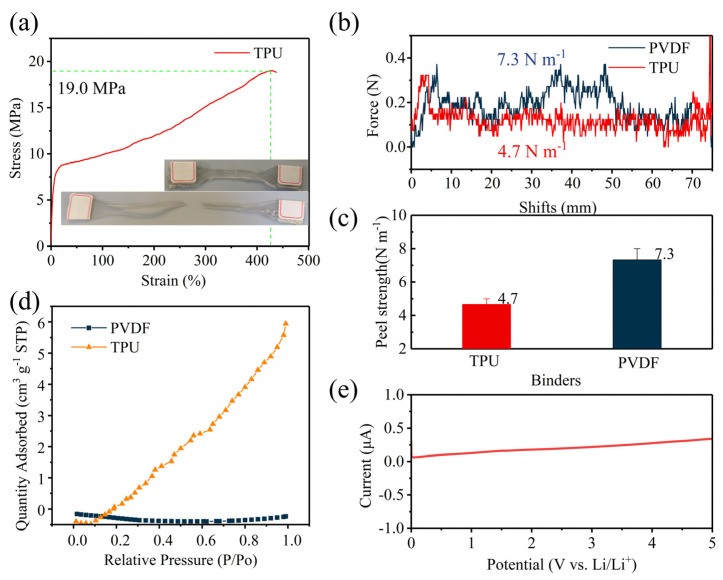
(**a**) Tensile strength–strain curves of TPU at 25 °C and 50 mm min^−1^; (**b**) 180° peel test of PVDF and TPU cathode at 25 °C and 300 mm min^−1^; (**c**) peel strength of PVDF and TPU; (**d**) CO_2_ absorption of PVDF and TPU powder tested with BET at 0 °C; (**e**) LSV curve of TPU in 0–5 V at 1 mV s^−1^.

**Figure 4 nanomaterials-14-01269-f004:**
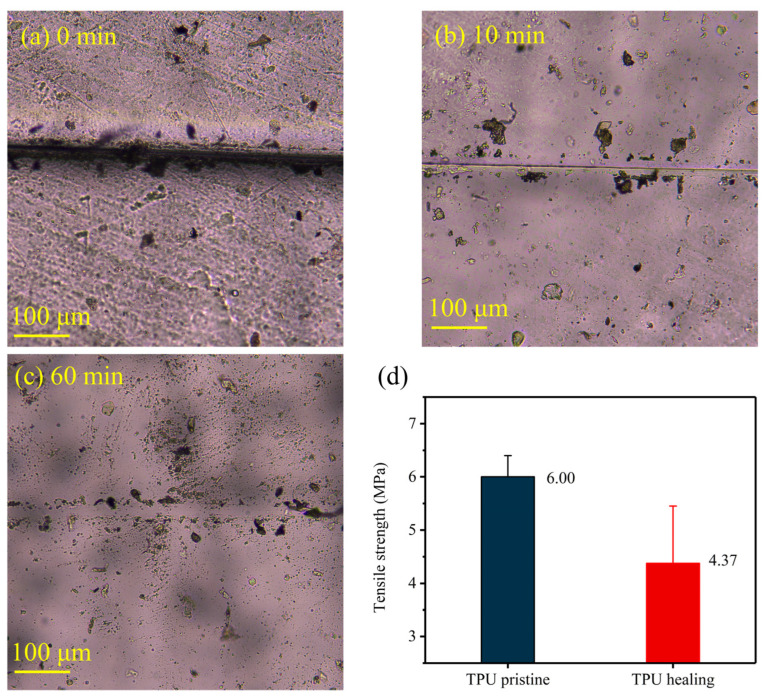
Optical microscopy images of TPU self-healed at 60 °C; the scratches healed gradually in (**a**) 0 min, (**b**)10 min, and (**c**) 60 min. (**d**) Tensile strength of pristine TPU and self-healed TPU.

**Figure 5 nanomaterials-14-01269-f005:**
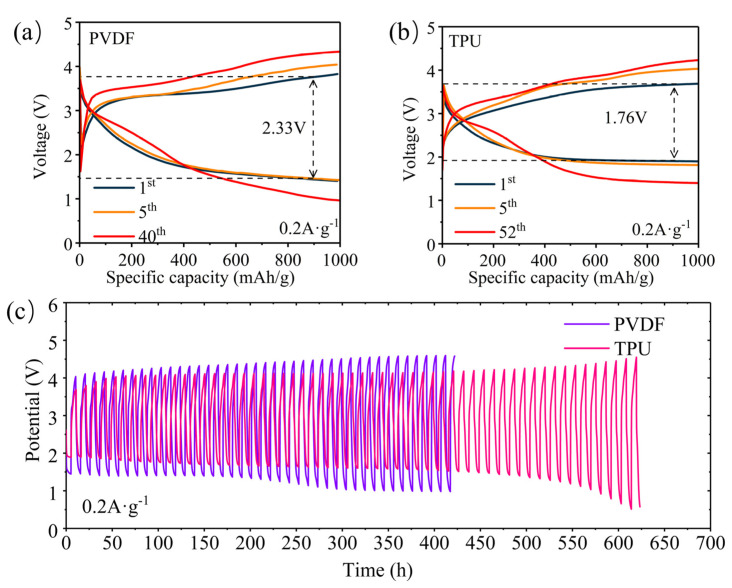
Voltage-specific capacity curves of Li-CO_2_ batteries at 0.2 A g^−1^ and 1000 mAh g^−1^ with (**a**) PVDF and (**b**) TPU binder. (**c**) The lifespan curves of Li-CO_2_ batteries at 0.2 A g^−1^ and 1000 mA hg^−1^ with PVDF and TPU binders.

**Figure 6 nanomaterials-14-01269-f006:**
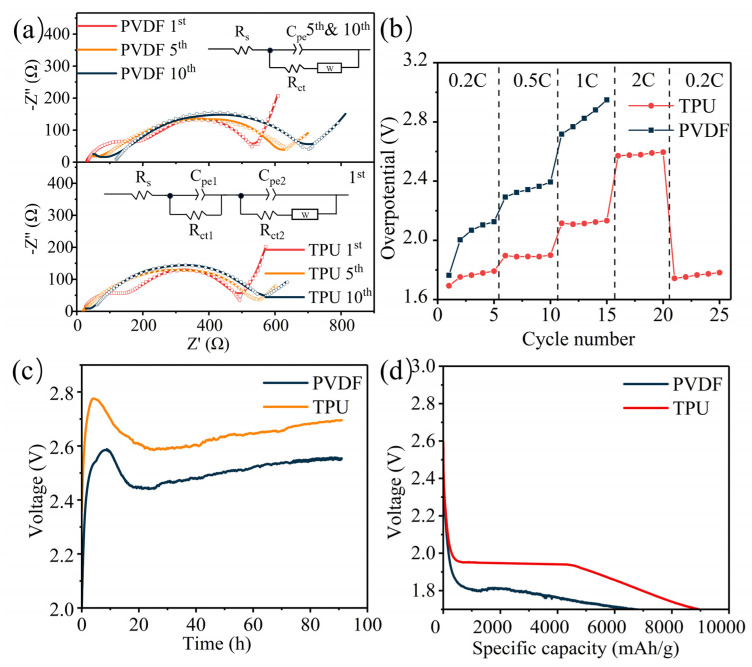
(**a**) Electrochemical impedance spectra (EIS) of Li-CO_2_ battery with PVDF and TPU after 1, 5, 10 cycles at the current density of 0.2 A g^−1^ and the cut-off capacity of 1000 mAh g^−1^. (**b**) Rate performance of Li-CO_2_ battery with TPU (red) and PVDF (blue) as binder at 0.2, 0.5, 1 and 2 C, the current density of 0.2 A g^−1^ and the cut-off capacity of 1000 mAh g^−1^. (**c**) The open voltage of Li-CO_2_ batteries with TPU (yellow) and PVDF (blue) statically placed under a CO_2_ atmosphere for 90 h. (**d**) The voltage-specific capacity profiles of a Li-CO_2_ battery with PVDF or TPU discharged to 1.7 V at 0.2 A g^−1^.

**Figure 7 nanomaterials-14-01269-f007:**
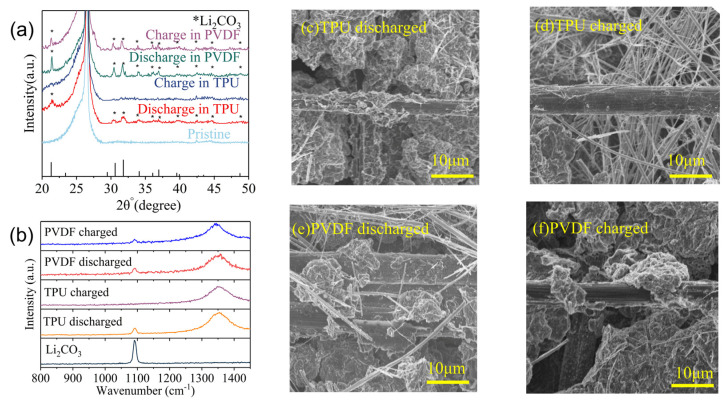
(**a**) XRD patterns of charged and discharged CO_2_ cathodes in Li-CO_2_ batteries with PVDF and TPU binders. (**b**) Raman spectra of charged and discharged CO_2_ cathodes in Li-CO_2_ batteries with PVDF and TPU binders. SEM images of (**c**) discharged Li-CO_2_ battery with TPU binders, (**d**) charged Li-CO_2_ battery with TPU binders, (**e**) discharged Li-CO_2_ battery with PVDF binders, and (**f**) charged Li-CO_2_ battery with PVDF binders after 30 cycles at 0.2 A g^−1^.

**Figure 8 nanomaterials-14-01269-f008:**
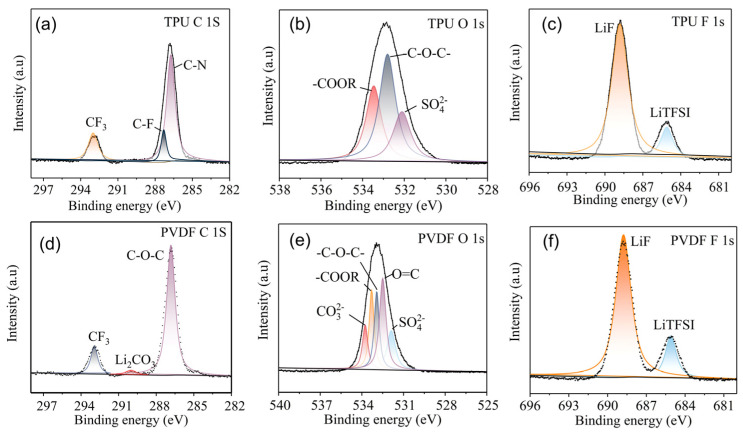
XPS spectra of (**a**) C(1s), (**b**) O(1s), and (**c**) F(1s) regions for Li metal of Li-CO_2_ batteries with TPU binders after 30 cycles at 0.2 A g^−1^; XPS spectra of (**d**) C(1s), (**e**) O(1s), and (**f**) F(1s) regions for Li metal of Li-CO_2_ batteries with PVDF binders after 30 cycles at 0.2 A g^−1^.

**Figure 9 nanomaterials-14-01269-f009:**
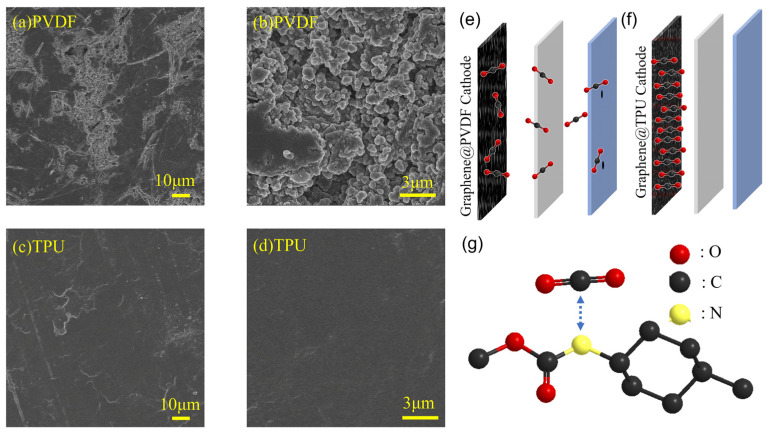
SEM images of anode of Li-CO_2_ batteries with PVDF at (**a**) 500× and (**b**) 5000× and TPU at (**c**) 500× and (**d**) 5000× after 30 cycles at 0.2 A g^−1^. (**e**) Schematic diagram of CO_2_ shuttle effect in Graphene@PVDF cathode. (**f**) Schematic diagram of CO_2_ shuttle effect reduced by Graphene@TPU cathode. (**g**) Schematic diagram of interaction between CO_2_ and -NH- of TPU.

## Data Availability

Data are available within the article.

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
