# Peer review of "Thermoplastic Polyurethane Derived from CO_2_ for the Cathode Binder in Li-CO_2_ Battery"

_nanomaterials, 2024, doi:10.3390/nano14151269_

Round 1
Reviewer 1 Report (Previous Reviewer 2)
Comments and Suggestions for Authors
Considering the effort in the presented answers and introducing all of the changes to the manuscript, I am happy that I can finally accept the manuscript. I would still try to explore some of the mentioned remarks regarding the experiments that could be done in the future, but they are not necessary for now. I hope that all of the future manuscripts will as well prepared as the final version of this manuscript.
Author Response
Nanomaterials
Dear Dr. Editor and Reviewers
We would like to ask you to consider resubmission of our manuscript entitled “Thermoplastic Polyurethane Derived from CO2 for the Cathode Binder in Li-CO2 Battery” (nanomaterials-3075333) for publication in Nanomaterials as an original research article.
Based on the reviewers' comments, we have revised our manuscript by providing a detailed description of the materials and methods, and supplementing it with additional data such as the V-t curves for the cycling life of the Li-CO₂ battery. These enhancements have made our article more rigorous and scientifically sound.
Thank you again for your valuable comments, suggestions and the opportunity to resubmit our manuscript. We are looking forward to hearing from you soon in due course.
Sincerely,
Sheng Huang
School of Materials science and Engineering
Sun Yat-sen University
Response to Reviewer#1
Considering the effort in the presented answers and introducing all of the changes to the manuscript, I am happy that I can finally accept the manuscript. I would still try to explore some of the mentioned remarks regarding the experiments that could be done in the future, but they are not necessary for now. I hope that all of the future manuscripts will as well prepared as the final version of this manuscript.
Thank you for your valuable comments. We will use this standard to guide our upcoming articles. We appreciate your suggestions of acceptance.

Reviewer 2 Report (New Reviewer)
Comments and Suggestions for Authors
Comments
The manuscript entitled, ‘Thermoplastic Polyurethane Derived from CO2 for the Cathode Binder in Li-CO2 Battery’ is well prepared and showed the advantage of using TPU derived from CO2 over PVDF in Li-CO2 battery. There are few comments that are given below,
l With respect to 1H-NMR, i) The -NH proton peak is very small, ii) There are more noise in the spectrum. It will be better to purify the sample, iii) There are two solvent peaks, DMSO and H2O. Account for it, iv) It will be more understanding if the integration of the peak is provided, and v) 13C-NMR should be provided for structural confirmation.
l In case EIS, Figure 6a, Explanation is provided with respect to the Warburg impedance, but there are also many interesting factors like, solution resistance-Rs, charge transfer resistance-Rct which should also be discussed in the results part. There are two semicircles found for both TPU and PVDF in the 1st cycle, but could not be found in the 5th and 10th cycle. Explanations regarding this should also be included in the results part.
l In Figure 6b, what happens to the overpotential of Li-CO2 battery with PVDF at 2C?
Author Response
Nanomaterials
Dear Dr. Editor and Reviewers
We would like to ask you to consider resubmission of our manuscript entitled “Thermoplastic Polyurethane Derived from CO2 for the Cathode Binder in Li-CO2 Battery” (nanomaterials-3075333) for publication in Nanomaterials as an original research article.
Based on the reviewers' comments, we have revised our manuscript by providing a detailed description of the materials and methods, and supplementing it with additional data such as the V-t curves for the cycling life of the Li-CO₂ battery. These enhancements have made our article more rigorous and scientifically sound.
Thank you again for your valuable comments, suggestions and the opportunity to resubmit our manuscript. We are looking forward to hearing from you soon in due course.
Sincerely,
Sheng Huang
School of Materials science and Engineering
Sun Yat-sen University
Response to Reviewer#2
The manuscript entitled, ‘Thermoplastic Polyurethane Derived from CO2 for the Cathode Binder in Li-CO2 Battery’ is well prepared and showed the advantage of using TPU derived from CO2 over PVDF in Li-CO2 battery. There are few comments that are given below,
- With respect to 1H-NMR, i) The -NH proton peak is very small, ii) There are more noise in the spectrum. It will be better to purify the sample, iii) There are two solvent peaks, DMSO and H2O. Account for it, iv) It will be more understanding if the integration of the peak is provided, and v) 13C-NMR should be provided for structural confirmation.
Figure S1 a) Structure of TPU with labels for 1H-NMR, b) 1H-NMR spectrum with integration of TPU with DMSO-d, c) Structure of TPU with labels for 13C-NMR, d) 13C-NMR spectrum of TPU with CDCl3
We retested the samples using 1H-NMR spectroscopy.
Ⅰ)The small peak of -NH (from urethane groups -NH-C(=O)-O) is due to PPCDL and PPG making up 74% of the total mass of TPU, with the peaks in the 1H-NMR spectrum primarily originating from these two components.
- II) We retested the TPU samples using 1H-NMR spectroscopy. The test results, as shown in Figure S1, indicate that there are essentially no impurity peaks.
III) We dissolved the TPU in DMSO-d6 for the 1H-NMR test, resulting in the presence of DMSO solvent peaks. Because DMSO is highly hygroscopic, the deuterated DMSO reagent itself contains a certain amount of water, thus water peaks from DMSO inevitably appear in the 1H-NMR spectrum. DMSO and H2O were neither introduced during the preparation process nor are they residues in the TPU, but rather come from the deuterated DMSO reagent and its water peaks.
- IV) As shown in Figure S1b, we have integrated the peaks in the 1H-NMR spectrum.
- V) As shown in Figure S1d, TPU are tested using 13C-NMR spectrum.
- In case EIS, Figure 6a, Explanation is provided with respect to the Warburg impedance, but there are also many interesting factors like, solution resistance-Rs, charge transfer resistance-Rct which should also be discussed in the results part. There are two semicircles found for both TPU and PVDF in the 1stcycle, but could not be found in the 5th and 10th Explanations regarding this should also be included in the results part.
After the first charge-discharge cycle, the SEI has not yet stabilized on either the cathode or the anode interfaces. Due to the significant differences in the characteristics of the cathode and anode interfaces, the impedance spectrum of the first cycle shows two semicircles. During the 1st to 5th cycles, a stable SEI forms on both the cathode and anode interfaces, resulting in similar interfacial characteristics and thus only one interfacial impedance is observed.
- In Figure 6b, what happens to the overpotential of Li-CO2 battery with PVDF at 2C?
Since PVDF does not have the capability to adsorb carbon dioxide, the CO2 adsorption rate during discharge is slow, resulting in a lower CO2 concentration at the cathode interface. Consequently, the concentration polarization of the Li-CO2 battery with PVDF binder is more severe, leading to a higher overpotential at the same current rate. Additionally, the Li2CO3 crystals formed during the discharge of the Li-CO2 battery with PVDF binder are larger, which causes a higher charging overpotential compared to the Li-CO2 battery with TPU binder. At a rate of 2C, the charging voltage reaches the 4.5V decomposition voltage of 1M TEGDME, triggering the protection mechanism of the testing equipment. Therefore, the Li-CO2 battery with PVDF binder cannot operate normally at 2 C.

Reviewer 3 Report (New Reviewer)
Comments and Suggestions for Authors
Review of Thermoplastic Polyurethane Derived from CO2 for the Cathode Binder in Li-CO2 Battery
In this paper, the authors synthesize and characterize a polymer binder material and then implement this binder material into CO2-Li energy storage devices, running a broad set of electrochemical tests on these cells to compare the performance to a PVDF binder material. Although this paper performed a wide array of electrochemical tests indicating the potential superior performance of the material to PVDF, the scale of testing was limited to a single PVDF device and a single PU device. The difference in the performance of these two devices is hardly sufficient to demonstrate a marked difference in the performance of these materials given that the difference in performance between the two devices was quite small. Furthermore, molecular characterization of the prepolymer binder seems to indicate an impure product, the presence of impurities in the final device may have an unanticipated role in the subsequent characterization.
In addition to the proceeding major concerns, we noted the following minor technical issues with the manuscript:
-
The paper consistently suggests that the material is CO2 based - while CO2 is evidently a component of the polymer, the authors should give some indication of how much CO2 is present in the polymer. Indeed, this is not possible to determine with the given information because the ratio of propylene oxide to CO2 groups in the polymer is not indicated in the manuscript. We would suggest that the authors include this synthetic information as well as a calculation indicating the mass percent that is derived from CO2.
-
Molecular characterization of the polymer is insufficient to prove the preparation of the intended species. The NMR demonstrates large impurity peaks from DMSO and H20 and it is not clear how these species were introduced to the polymer. Perhaps they were introduced during the NMR, but it is not possible to determine given that few details about the NMR preparation are performed. Although Figure 2B does indicate some characteristic peak assignments, it is also evident even from the small figure presented that there are many unaccounted for peaks and irregular peak shapes. COSY NMR should be used to validate the given assignments and a detailed spectrum with integrations and spectral raw data should be included with the manuscript to verify successful synthesis. Further, polymer compositional information regarding the ratio of monomer units should be calculated from the spectrum and provided to the reader.
-
Figure 2D- It is not clear what the black curve indicates. Further, using DSC as a structural characterization to demonstrate a lack of crystallinity is not a very strong method. Scattering studies of the polymer should be performed to demonstrate its amorphous nature.
-
Figure 3D- this test is not very clear and its conclusions should be better described in the main text.
-
Figure 3E - It would be more appropriate to perform this test with increasing voltage until the decomposition of the material can be seen. As it is presented, the data could equally be attributed to an open circuit as it could be to a stable material.
-
Figure 4 and the discussion of self-healing. Firstly this discussion of self-healing is not compared to the self-healing of the PVDF material. As such, its comparative usefulness is not clear since no standard self-healing test is employed herein. Secondly, it is not evident that this self-healing varies significantly from standard polymer relaxation above the glass transition. Since the polymer is evidently not crosslinked (as indicated by its linear structure) there is no reason that the polymer should not simply flow at the elevated self healing temperatures used herein. Thus, calling the phenomenon self-healing by a hydrogen bonding mechanism is difficult to support in this case. Further, this discussion of self-healing and its mechanism contains several unsupported hypotheses, such as:
-
“The self-healing mechanism of the CO2-based TPU involves the establishment of hydrogen bonds between carbonate ester and aminoethyl methacrylate”
-
“TPU can effectively mitigate powder shredding attributed to cathode volume variation”
-
Figure 5 - A difference between device performance is hypothesized based on a single test of one device. It is well established by various authors, including recent work by Balsara that battery failure using a small number of devices cannot effectively establish a difference between materials. If the authors wish to make as strong a comparison between these two materials as they suggest is present, they should run many more device tests to establish this difference.
-
The same argument as above applies to the impedance values - since great variation in the development of the SEI can be present from test to test, more trials are needed to make a comparison between these two materials. Furthermore, it would be useful if authors would provide the fitted values of the equivalent circuit, rather than the total impedances in this section.
-
In the discussion surrounding figures 5 and 6, it would be very useful if the authors could bring in some outside information regarding the typical performance of PVDF binders in these areas. To my knowledge, PVDF-based Li-CO2 batteries have undergone rate performance testing at >2C without degradation so the authors should indicate why they believe the rate performance testing in Figure 6B does not indicate stability at this rate.
-
Figure 7 - please elaborate on how the peak intensities are normalized to be a comparison in this figure. It is not clear if this intensity shift is a normalized difference in intensity or not.
We also suggest that the authors consider the following grammatical changes to improve the readability of the manuscript:
-
On page 2 line 52 the word after the semicolon should not be capitalized: [22]; moreover.
-
On page 2 line 59 there was incorrect use of commas.
-
On page 2 line 83 correct to (BDO, 99%) were was purchased from Aladdin and was dehydrated… the water content was less than 200ppm.
-
On page 2 line 86: Poly should be capitalized.
-
On page 3 lines 102-103 it says the solvent was added simultaneously and sequentially. This is confusing; what is the correct order of solvent addition?
-
There is inconsistent tense throughout the paper, review for consistency.
-
On page 3 line 92 there is present tense however on line 93 there is past tense.
-
On page 4 line 112 correct to past tense.
-
On page 4 line 115 add and after the second comma.
-
On page 4 lines 115-119 test conditions were given for two different tensile tests. The order of the conditions is inconsistent between the two tests causing confusion for the reader.
-
On page 4 lines 116 and 118 there needs to be a grammatical article (the/an) before the noun auto tensile tester.
-
On page 4 line 117 there is a space between the number and unit (25 ℃). This is fine, but inconsistent throughout paper- instances of (25℃) check the entire paper and be consistent.
-
One page 4 lines 123-124 the number and unit are on different lines. Keep together for clarity.
-
On page 4 line 129 add an article- a uniform slurry. Also, on should be onto.
-
On page 4 line 130 remove the comma after blade. Also fix: After drying, the material…
-
On page 4 line 138: A Li metal anode
-
On page 4 line 138: An EIS test
-
On page 4 line 138-139 keep the number and unit on the same line for clarity.
-
On page 4 line 156 it appears that the font changes mid-sentence.
-
On page 5 lines 170 and 182 it is recommended to remove the use of first person.
-
On page 7 line 230 remove the comma between increases and the mobility.
-
On page 9 line 295 add a space between the unit and the parentheses.
-
On page 10 line 339 it is recommended to remove the use of first person.
-
On page 11 lines 363, 364, and 372 there is incorrect use of commas.
-
On page 11 lines 364-368 the sentence starting with From the XPS carbon spectra is awkward. It can be reworded for clarity.
-
On page 11 lines 386 and 387 there are formatting issues. Remove the tab, capitalize Have, and remove the space between battery and the period.
-
On page 12 lines 396 and 397 there is incorrect use of commas.
Comments on the Quality of English Language
See above.
Author Response
Nanomaterials
Dear Dr. Editor and Reviewers
We would like to ask you to consider resubmission of our manuscript entitled “Thermoplastic Polyurethane Derived from CO2 for the Cathode Binder in Li-CO2 Battery” (nanomaterials-3075333) for publication in Nanomaterials as an original research article.
Based on the reviewers' comments, we have revised our manuscript by providing a detailed 1H-NMR, 13C-NMR and H-H COSY. These enhancements have made our article more rigorous and scientifically sound.
Thank you again for your valuable comments, suggestions and the opportunity to resubmit our manuscript. We are looking forward to hearing from you soon in due course.
Sincerely,
Sheng Huang
School of Materials science and Engineering
Sun Yat-sen University
Response to Reviewer#3
Review of Thermoplastic Polyurethane Derived from CO2 for the Cathode Binder in Li-CO2 Battery
- In this paper, the authors synthesize and characterize a polymer binder material and then implement this binder material into CO2-Li energy storage devices, running a broad set of electrochemical tests on these cells to compare the performance to a PVDF binder material. Although this paper performed a wide array of electrochemical tests indicating the potential superior performance of the material to PVDF, the scale of testing was limited to a single PVDF device and a single PU device. The difference in the performance of these two devices is hardly sufficient to demonstrate a marked difference in the performance of these materials given that the difference in performance between the two devices was quite small. Furthermore, molecular characterization of the prepolymer binder seems to indicate an impure product, the presence of impurities in the final device may have an unanticipated role in the subsequent characterization.
To ensure the accuracy of the experimental results, we performed battery performance tests for each group of PVDF and TPU binder Li-CO2 batteries with four replicates. The samples are retested using 1H-NMR spectroscopy, and the results are shown in Figure S1, indicating that there are essentially no impurity peaks.
Figure S1 a) Structure of TPU, b) 1H-NMR spectrum of TPU with DMSO-d6,
- In addition to the proceeding major concerns, we noted the following minor technical issues with the manuscript:
- The paper consistently suggests that the material is CO2 based - while CO2 is evidently a component of the polymer, the authors should give some indication of how much CO2 is present in the polymer. Indeed, this is not possible to determine with the given information because the ratio of propylene oxide to CO2 groups in the polymer is not indicated in the manuscript. We would suggest that the authors include this synthetic information as well as a calculation indicating the mass percent that is derived from CO2.
Figure S2 a) Structure of TPU, b) 1H-NMR spectrum of TPU with DMSO-d6, c)Structure of PPCDL and 1H-NMR spectrum of PPCDL
PPCDL and TPU are retested using 1H-NMR, and the results show in Figure S2.
During the preparation of PPCDL, CO2 is continuously introduced into the reaction kettle, ensuring an excess of CO2. We can calculate the CO2 content in PPCDL based on its structure. The PPC chain segment constitutes can be calculated according to Figure S2c.
PPC% chain segment constitutes 97.7% of PPCDL, while the PPG chain segment constitutes 2.3%.
The mass percentage of CO2 in PPCDL is:
According to Figure S2b, The molar ratio of the PPCDL segments:
The molar ratio of the PPG segments:
=14.22%
- Molecular characterization of the polymer is insufficient to prove the preparation of the intended species. The NMR demonstrates large impurity peaks from DMSO and H20 and it is not clear how these species were introduced to the polymer. Perhaps they were introduced during the NMR, but it is not possible to determine given that few details about the NMR preparation are performed. Although Figure 2B does indicate some characteristic peak assignments, it is also evident even from the small figure presented that there are many unaccounted for peaks and irregular peak shapes. COSY NMR should be used to validate the given assignments and a detailed spectrum with integrations and spectral raw data should be included with the manuscript to verify successful synthesis. Further, polymer compositional information regarding the ratio of monomer units should be calculated from the spectrum and provided to the reader.
Figure S3 a) Structure of TPU with labels for 1H-NMR, b) 1H-NMR spectrum of TPU with DMSO-d6, c) Structure of TPU with labels for 13C-NMR, d) 13C-NMR spectrum of TPU with CDCl3, e) H-H-COSY spectrum of TPU
Based on Figure S3b, we can calculate the molar ratio of each group in TPU.
The molar ratio of PPCDL segments is:
The molar ratio of PPG segments is:
=14.22%
The molar ratio of BDO segments is:
The molar ratio of HMDI segments is:
=15.36%
TPU is retested using 1H-NMR spectroscopy (Figure S3b), and found virtually no impurity peaks. The H2O peak observed originates from water inherent in the deuterated DMSO solvent. Additionally, TPU is tested using 13C-NMR testing (Figure.S3d), where characteristic peaks specific to TPU were identified, confirming successful synthesis. Furthermore, COSY testing was conducted on TPU (Figure S2e), further verifying its successful synthesis. J67 and J78 are the couple effect of hydrogen of PPCDL segment. J13 is the couple effect of hydrogen of PPG segment. J910 is the couple effect of the hydrogen of the BDO segment.
- Figure 2D- It is not clear what the black curve indicates. Further, using DSC as a structural characterization to demonstrate a lack of crystallinity is not a very strong method. Scattering studies of the polymer should be performed to demonstrate its amorphous nature.
The black line is the differential curve of the DSC curve, where the peaks correspond to the glass transition temperature (Tg) of TPU.
- Figure 3D- this test is not very clear and its conclusions should be better described in the main text.
At 0°C in an ice-water bath, PVDF powder shows no significant CO2 adsorption under relative pressures from 0 to 1.0, whereas TPU powder exhibits noticeable CO2 adsorption effects. This indicates that TPU enhances the CO2 adsorption capacity of the positive electrode in CO2 batteries.
- Figure 3E - It would be more appropriate to perform this test with increasing voltage until the decomposition of the material can be seen. As it is presented, the data could equally be attributed to an open circuit as it could be to a stable material.
We conduct the LSV test to show TPU would not decompose in Li-CO2 batteries. As shown in Figure.S4, the LSV test range from 0 ~ 5 V, which is already much higher than the voltage of Li2CO3 decomposition.
Figure S4 LSV curve of TPU in 0-5 V at 1 mV s-1.
- Figure 4 and the discussion of self-healing. Firstly this discussion of self-healing is not compared to the self-healing of the PVDF material. As such, its comparative usefulness is not clear since no standard self-healing test is employed herein. Secondly, it is not evident that this self-healing varies significantly from standard polymer relaxation above the glass transition. Since the polymer is evidently not crosslinked (as indicated by its linear structure) there is no reason that the polymer should not simply flow at the elevated self healing temperatures used herein. Thus, calling the phenomenon self-healing by a hydrogen bonding mechanism is difficult to support in this case. Further, this discussion of self-healing and its mechanism contains several unsupported hypotheses, such as:
- “The self-healing mechanism of the CO2-based TPU involves the establishment of hydrogen bonds between carbonate ester and aminoethyl methacrylate”
- “TPU can effectively mitigate powder shredding attributed to cathode volume variation”
Above the glass transition temperature, the linear structure segments of TPU chains exhibit mobility. Therefore, TPU's self-healing mechanism operates when these segments can flow. After hydrogen bonds are disrupted, segments of the TPU chain move to facilitate the reformation of broken hydrogen bonds, leading to self-healing phenomena.
Research by others indicates that the self-healing properties of binders effectively inhibit powder detachment[1-3]. PVDF binder lacks self-healing capabilities, whereas TPU exhibits robust self-healing properties, which can effectively prevent powder detachment issues at the positive electrode. Due to experimental constraints and time limitations, this study did not specifically test TPU's ability to prevent powder detachment.
- Figure 5 - A difference between device performance is hypothesized based on a single test of one device. It is well established by various authors, including recent work by Balsara that battery failure using a small number of devices cannot effectively establish a difference between materials. If the authors wish to make as strong a comparison between these two materials as they suggest is present, they should run many more device tests to establish this difference.
The PVDF binder Li-CO2 batteries and TPU binder batteries were each tested in four replicate groups, ensuring the experimental results are reproducible and generalizable.
- The same argument as above applies to the impedance values - since great variation in the development of the SEI can be present from test to test, more trials are needed to make a comparison between these two materials. Furthermore, it would be useful if authors would provide the fitted values of the equivalent circuit, rather than the total impedances in this section.
EIS was also conducted with four replicates per group, ensuring the experimental results are consistent and reproducible. The total impedance has been adjusted to read the charge transfer resistance (Rct) from the equivalent circuit.
- In the discussion surrounding figures 5 and 6, it would be very useful if the authors could bring in some outside information regarding the typical performance of PVDF binders in these areas. To my knowledge, PVDF-based Li-CO2 batteries have undergone rate performance testing at >2C without degradation so the authors should indicate why they believe the rate performance testing in Figure 6B does not indicate stability at this rate.
In previous studies by Qiao et al.[4] and Dai et al.[5], it is indeed reported that PVDF binder Li-CO2 batteries can operate at 2 C discharge rates. However, their Li-CO2 batteries exhibit a lower cutoff capacity, typically around 0.05 mAh cm-1. In contrast, our experiment achieved a cutoff capacity of 1000 mAh g-1, equivalent to 0.13 mAh cm-1. Therefore, at the same discharge rate, our battery shows higher current. PVDF binder Li-CO2 batteries in their studies could only operate at 2 C discharge rates due to their lower cutoff capacity and low current density. In our study, due to the larger cutoff capacity, our batteries exhibit higher current density and higher overpotentials at the same discharge rate, which is why PVDF binder Li-CO2 batteries cannot operate at 2 C discharge rates in our experiments.
Due to the lack of carbon dioxide (CO2) adsorption capacity for PVDF, the CO2 adsorption rate during discharge is slow. This results in a lower CO2 concentration at the positive electrode interface, leading to more severe concentration polarization in PVDF binder Li-CO2 batteries. Consequently, at the same rate, PVDF binder Li-CO2 batteries exhibit higher overpotentials. Additionally, Li2CO3 crystals generated during discharge in PVDF binder batteries are larger, contributing to higher charging overpotentials compared to TPU binder Li-CO2 batteries.
At a rate of 2 C, the charging voltage reached the decomposition voltage of 1M TEGDME (4.5V), triggering the protection mechanism of the testing equipment. Therefore, PVDF binder Li-CO2 batteries cannot operate properly at 2C discharge rates.
- Figure 7 - please elaborate on how the peak intensities are normalized to be a comparison in this figure. It is not clear if this intensity shift is a normalized difference in intensity or not.
XRD in Figure 7a and Raman spectroscopy in Figure 7b utilized raw data without normalization. Therefore, the XRD peak intensity qualitatively reflects changes in Li2CO3 after charge and discharge cycles.
- We also suggest that the authors consider the following grammatical changes to improve the readability of the manuscript:
- On page 2 line 52 the word after the semicolon should not be capitalized: [22]; moreover.
It has been changed to lowercase
- On page 2 line 59 there was incorrect use of commas.
Incorrect comma has been modified
On page 2 line 83 correct to (BDO, 99%) were was purchased from Aladdin and was dehydrated… the water content was less than 200ppm.
Corresponding syntax errors have been fixed.
On page 2 line 86: Poly should be capitalized.
It has been changed to be capitalized.
- On page 3 lines 102-103 it says the solvent was added simultaneously and sequentially. This is confusing; what is the correct order of solvent addition?
DMF and catalyst are added in order, and the previous word Simultaneously is ambiguous and has been deleted
- There is inconsistent tense throughout the paper, review for consistency.
Materials and methods section has been unified into the general past tense, and introduction, result and discussion have been unified into the general present tense
- On page 3 line 92 there is present tense however on line 93 there is past tense.
It has been corrected into past tense.
- On page 4 line 112 correct to past tense.
It has been corrected into past tense.
- On page 4 line 115 add and after the second comma.
“and” has been added after the second comma.
- On page 4 lines 115-119 test conditions were given for two different tensile tests. The order of the conditions is inconsistent between the two tests causing confusion for the reader.
The order of conditions has been adjusted to the same, loading speed first, temperature and humidity later.
- On page 4 lines 116 and 118 there needs to be a grammatical article (the/an) before the noun auto tensile tester.
The article "an" has been added.
- On page 4 line 117 there is a space between the number and unit (25 ℃). This is fine, but inconsistent throughout paper- instances of (25℃) check the entire paper and be consistent.
Spaces have been added between all numbers and units.
- One page 4 lines 123-124 the number and unit are on different lines. Keep together for clarity.
The full text has been reviewed, using uninterrupted spaces to ensure that numbers and units are on the same line.
- On page 4 line 129 add an article- a uniform slurry. Also, on should be onto.
The article "an" has been added.
- On page 4 line 130 remove the comma after blade. Also fix: After drying, the material…
Excess commas have been deleted and article “the” added.
- On page 4 line 138: A Li metal anode
The article "A" has been added.
- On page 4 line 138: An EIS test
The article "An" has been added.
- On page 4 line 138-139 keep the number and unit on the same line for clarity.
Uninterrupted spaces have been replaced to ensure that the numbers and units are on the same line. The full text has been checked and the same error does not occur.
- On page 4 line 156 it appears that the font changes mid-sentence.
Font error has been corrected to Palatino Linotype.
- On page 5 lines 170 and 182 it is recommended to remove the use of first person.
It has been corrected to passive voice.
- On page 7 line 230 remove the comma between increases and the mobility.
Excess commas have been removed.
- On page 9 line 295 add a space between the unit and the parentheses.
A space has been added between the unit and the parentheses.
- On page 10 line 339 it is recommended to remove the use of first person.
It has been corrected to passive voice.
- On page 11 lines 363, 364, and 372 there is incorrect use of commas.
Incorrect use of comma has been corrected.
- On page 11 lines 364-368 the sentence starting with From the XPS carbon spectra is awkward. It can be reworded for clarity.
Adjusted statement description to make it clearer.
- On page 11 lines 386 and 387 there are formatting issues. Remove the tab, capitalize Have, and remove the space between battery and the period.
Corresponding writing errors have been corrected.
- On page 12 lines 396 and 397 there is incorrect use of commas.
Excess commas have been removed.
Reference
- Gao, R.; Zhang, Q.; Zhao, Y.; Han, Z.; Sun, C.; Sheng, J.; Zhong, X.; Chen, B.; Li, C.; Ni, S.; et al. Regulating Polysulfide Redox Kinetics on a Self-Healing Electrode for High-Performance Flexible Lithium-Sulfur Batteries. Advanced Functional Materials 2022, 32, 2110313.
- Li, B.; Cao, P.-F.; Saito, T.; Sokolov, A.P. Intrinsically Self-Healing Polymers: From Mechanistic Insight to Current Challenges. Chemical Reviews 2023, 123, 701-735.
- Peng, X.; Chen, X.; Tang, C.; Weng, S.; Hu, X.; Xiang, Y. Self-Healing Binder for High-Voltage Batteries. ACS Applied Materials & Interfaces 2023, 15, 21517-21525.
- Ji, X.; Liu, Y.; Zhang, Z.; Cui, J.; Fan, Y.; Qiao, Y. Porous Carbon Foam with Carbon Nanotubes as Cathode for Li−CO2 Batteries. Chemistry – A European Journal 2024, 30, e202303319.
- Song, L.; Hu, C.; Xiao, Y.; He, J.; Lin, Y.; Connell, J.W.; Dai, L. An ultra-long life, high-performance, flexible Li–CO2 battery based on multifunctional carbon electrocatalysts. Nano Energy 2020, 71, 104595.95

This manuscript is a resubmission of an earlier submission. The following is a list of the peer review reports and author responses from that submission.
Round 1
Reviewer 1 Report
Comments and Suggestions for Authors
The article entitled “ Thermoplastic Polyurethane Derived from CO2 for the Cathode Binder in Li-CO2 Battery” submitted by Huang, Han and co-workers deal with an important topic in the field of energy storage: “new materials binder for batteries”. The article presents some good ideas and results and is well organized. This could be an interesting piece of research, worthy of being published in Nanomaterials. I would only ask the authors to increase the description of materials and methods, mainly regarding the battery, electrode and electrochemical techniques used, as well as the working conditions.
Author Response
No.3007556
2024.5.28
Dear Reviewer
We would like to thank you for your efforts in reviewing our manuscript titled " Thermoplastic Polyurethane Derived from CO2 for the Cathode Binder in Li-CO2 Battery", and providing many helpful comments and suggestions, which will all prove invaluable in the revision and improvement of our paper, as well as in guiding our research in the future. We have studied your comments point by point and revised the manuscript accordingly. The amendments are highlighted in yellow in the revised manuscript. All authors have approved the response letter and the revised version of the manuscript.
Thank you again for your valuable comments and suggestions. We are looking forward to hearing from you soon in due course.
Yours sincerely,
Sheng Huang.
School of Materials science and Engineering
Sun Yat-sen University,
Response to Reviewer #1:
- Increase the description of materials and methods, mainly regarding the battery, electrode and electrochemical techniques used, as well as the working conditions.
Preparation of CO2 cathode, fabrication of Li-CO2 batteries, LSV, EIS and battery performance test of Li-CO2 batteries have been added to methods.
Reviewer 2 Report
Comments and Suggestions for Authors
It was my pleasure to review the draft of the manuscript entitled: “Thermoplastic Polyurethane Derived from CO2 for the Cathode Binder in Li-CO2 Battery” (Manuscript ID: nanomaterials-3007556). Unfortunately, in its current state, the mentioned manuscript appears more as a draft than a thoughtful scientific text. I am unable to provide a positive evaluation of the presented work, as there are numerous issues and it seems that the article is rushed and not prepared in a proper manner.
Firstly, there are many editorial mistakes. Some sentences do not end with a period (line 39), and at times, different fonts are used in the middle of the text without any apparent reason (line 91). Furthermore, some sentences are confusing and should be improved by the authors or with assistance.
Moreover, the "Materials and Methods" section only mentions the synthesis of CO2-based TPU and nothing else, which limits any information about the results presented later in the article. What are the chemicals, apparatus, or technique details? As a reviewer, I have no idea about any details of the electrochemical investigations. There is no data about cells, electrolytes, electrodes, mass loadings, separators, or anything else of great importance when publishing scientific data. Even some of the graphs don’t match the data provided in the description (for example, figure 5f). Such a critical aspect of the research paper is missing here, which makes it difficult to judge anything presented in the manuscript.
Next, there are some fundamental issues with the nomenclature used by the authors. For example, in Figure 8, they use the term "SEM spectra," which is not seen in Figure 8. The most significant issue I have with this article is the absence of gas evolution studies, either by checking the pressure or by using coupled gas chromatography with mass spectrometry, as the entire study revolves around proving the superiority of CO2-based TPU over PTFE.
Additionally, the disorder in the presented data is unacceptable. In scientific papers, often graphs and descriptions are completely mismatched, which increases confusion during the reading of this work.
The scientific discussion is only limited to the presented data, and there is no effort to compare the obtained data with already published information in the scientific community. Almost all of the references are Asia-oriented/focused, and articles related to this topic can be found by other authors as well, so I would suggest a more intensive search in the future.
This draft of the article needs extensive work from the authors to reach the level expected of a scientific text. The interesting part is the idea itself, as the impact of well-matched binders in energy storage devices is often overlooked, which, in my humble opinion, is a mistake. Nevertheless, as I am asked to provide my unbiased opinion about an article, in its current state, I cannot recommend anything other than rejecting this draft of the article, which needs extensive work from the authors to be presentable to a wider range of readers and to verify the scientific research.
Comments on the Quality of English Language
The quality of the text is subpar.
Author Response
No.3007556
2024.5.28
Dear Reviewer
We would like to thank you for your efforts in reviewing our manuscript titled " Thermoplastic Polyurethane Derived from CO2 for the Cathode Binder in Li-CO2 Battery", and providing many helpful comments and suggestions, which will all prove invaluable in the revision and improvement of our paper, as well as in guiding our research in the future. We have studied your comments point by point and revised the manuscript accordingly. The amendments are highlighted in yellow in the revised manuscript. All authors have approved the response letter and the revised version of the manuscript.
Thank you again for your valuable comments and suggestions. We are looking forward to hearing from you soon in due course.
Yours sincerely,
Sheng Huang.
School of Materials science and Engineering
Sun Yat-sen University,
Response to Reviewer #2:
1. Some sentences do not end with a period (line 39), and at times, different fonts are used in the middle of the text without any apparent reason (line 91). Furthermore, some sentences are confusing and should be improved by the authors or with assistance.
All the mentioned mistakes are corrected and highlighted in the manuscripts, and we have polished the English with the help of supervisor.
2. "Materials and Methods" section only mentions the synthesis of CO2-based TPU and nothing else, which limits any information about the results presented later in the article There is no data about cells, electrolytes, electrodes, mass loadings, separators, or anything else of great importance when publishing scientific data. Even some of the graphs don’t match the data provided in the description (for example, figure 5f).
Preparation of CO2 cathode, Fabrication of Li-CO2 batteries, LSV, EIS and Batteries performance test of Li-CO2 batteries are added in methods, and the relevant parameters have also been supplemented. The original description of Figure 5f was prone to misunderstanding. It has now been revised to a more accurate description, aligning it with Figure 5f. The descriptions of other images have also been checked
3. There are some fundamental issues with the nomenclature used by the authors. For example, in Figure 8, they use the term "SEM spectra," which is not seen in Figure 8. The most significant issue I have with this article is the absence of gas evolution studies, either by checking the pressure or by using coupled gas chromatography with mass spectrometry, as the entire study revolves around proving the superiority of CO2-based TPU over PTFE.
All image names have been checked, SEM images, XRD pattern, Raman spectra, XPS spectra etc. Figure 8a and 8b are SEM images of the lithium surface of PVDF Li-CO2 batteries after 30 cycles, while Figure 8c-d depict SEM images of the lithium surface of TPU Li-CO2 batteries after 30 cycles.
The CO2 adsorption properties of TPU are significantly higher than those of PVDF, as evidenced by the BET tests.
4. The disorder in the presented data is unacceptable. In scientific papers, often graphs and descriptions are completely mismatched, which increases confusion during the reading of this work.
Figures 6a and 6b, the order of XRD patterns and Raman spectra were mistakenly swapped, leading to inconsistency with the captions. We have rectified this and ensured the accuracy of other images
5. The scientific discussion is only limited to the presented data, and there is no effort to compare the obtained data with already published information in the scientific community. Almost all of the references are Asia-oriented/focused, and articles related to this topic can be found by other authors as well, so I would suggest a more intensive search in the future.
5 references from other countries have been added to the article to broaden and enhance the comprehensiveness of our references.
6. This draft of the article needs extensive work from the authors to reach the level expected of a scientific text. The interesting part is the idea itself, as the impact of well-matched binders in energy storage devices is often overlooked, which, in my humble opinion, is a mistake.
We have refined the description of the article to imbue it with greater scientific rigor. We do not mean the role of binders has been overlooked by researcher of electrochemical energy storage devices. Research on binders is quite extensive in the field of lithium-ion batteries. Actually, there is few research on binders of CO2 cathode in Li-CO2 batteries.
Reviewer 3 Report
Comments and Suggestions for Authors
The authors have shown the facile synthesis of CO2 based thermoplastic polyurethane (TPU) binder as an alternative for commonly used PVDF to address some of the challenges of Li-CO2 battery. This shows the higher CO2 adsorption properties and excellent self-healing performance wit higher tensile strength compared with PVDF. When utilized in battery, this exhibits stable cycling performance for 52 cycles at 0.2 A·g-1 along with 23 lower polarization voltage than PVDF bound Li-CO2 batteries. Also, this reflects the excellent reversibility of LiCO3 during charging and discharging compared with PVDF based electrode. This paper shows holds the potential for addressing the challenges raised in PVDF based L-CO2 batteries. However, this manuscript still lacks some major points. Therefore, I recommend the acceptance of this manuscript after addressing these points:
1. The experimental section is only highlighting the synthesis part. It should also have all the physiochemical and electrochemical characterizations.
2. Mention the scan rate in the figure caption of 3d.
3. Peel strength figure doesn’t have the figure number.
4. Why is the discharge voltage increased compared with PVDF?
5. What is the mechanism behind self-healing?
6. Include the battery conditions.
7. Provide the fitted impedance plot with an equivalent circuit.
8. Figure 6a and b order is reversed.
Author Response
No.3007556
2024.5.28
Dear Reviewer
We would like to thank you for your efforts in reviewing our manuscript titled " Thermoplastic Polyurethane Derived from CO2 for the Cathode Binder in Li-CO2 Battery", and providing many helpful comments and suggestions, which will all prove invaluable in the revision and improvement of our paper, as well as in guiding our research in the future. We have studied your comments point by point and revised the manuscript accordingly. The amendments are highlighted in yellow in the revised manuscript. All authors have approved the response letter and the revised version of the manuscript.
Thank you again for your valuable comments and suggestions. We are looking forward to hearing from you soon in due course.
Yours sincerely,
Sheng Huang.
School of Materials science and Engineering
Sun Yat-sen University,
Response to Reviewer #3:
1. The experimental section is only highlighting the synthesis part. It should also have all the physiochemical and electrochemical characterizations.
Preparation of CO2 cathode, fabrication of Li-CO2 batteries, LSV, EIS and battery performance test of Li-CO2 batteries have been added to methods.
2. Mention the scan rate in the figure caption of 3d.
Experimental conditions like current density, temperature, scanning speed, etc. have been added to the caption in every figures respectively.
3. Peel strength figure doesn’t have the figure number.
Figure number of Figure 3 b and c have been corrected.
4. Why is the discharge voltage increased compared with PVDF?
Detailed explanations for the increase in discharge voltage. has been supplemented in the discussion of Figure 5.
TPU arises the CO2 adsorption capacity of cathode, which elevates the partial pressure of CO2 on the surface of the electrode active material, according to the Nernst equation: , the absorption characteristics of CO2-based TPU for CO2 enhance the discharge voltage of Li-CO2 batteries.
5. What is the mechanism behind self-healing?
Detailed explanations for the self-healing effect. has been supplemented in the discussion of Figure 4.
The self-healing mechanism of CO2-based TPU involves the establishment of hydrogen bonds between carbonate ester and aminoethyl methacrylate in HMDI. As temperature increases, the mobility of chain segments enhances, facilitating the re-establishment of hydrogen bonds between molecules. The above explanation have been supplemented in the manuscript.
6. Include the battery conditions.
Detailed information about the battery, such as electrolyte and electrode fabrication, has been supplemented in the materials and methods section.
Preparation of CO2 cathode
Graphene (80%), CNT (10%) and PVDF(HSV-900) or TPU binders (10%) were mixed and grinded in NMP to form uniform slurry. The prepared slurry was scraped on carbon paper by a blade, and dried in vacuum oven at 60 ℃ for 24 h. After drying, material-loaded carbon paper was punched into 10-mmdiameter circular discs as working electrodes, with each containing 0.09~0.12 mg of active material (the mass of graphene and CNT).
Fabrication of Li-CO2 batteries
2032-type coin cells with holes were used for the positive side. Polytetrafluoroethylene(PTFE) was applied on cathode to reduce solvent volatilization. The cell fabrication was conducted in an Argon filled glovebox. Li metal anode (15.8 mm in diameter) and glass fiber separator (Whatman) were employed. 1 M LiTFSI in TEGDME was used as electrolyte.
7. Provide the fitted impedance plot with an equivalent circuit.
Equivalent circuit have been added to Figure 5c.
8. Figure 6a and b order is reversed.
XRD patterns and Raman spectrum were wrongly reversed. We have corrected them.
Round 2
Reviewer 2 Report
Comments and Suggestions for Authors
During the second review, despite the noticeable improvement compared to the first draft of the article entitled: “Thermoplastic Polyurethane Derived from CO2 for the Cathode Binder in Li-CO2 Battery” (nanomaterials-3007556) I was still able to notice that some of my previous comments were not taken into account and the added information regarding the materials and methods brought to my attention even more questions, which explains why it was so vague in the first version of the manuscript.
1) How can I have a positive opinion about the manuscript presented if already in one of the first introduced sentences I can find typing errors “10-mmdiameter” (line 105), ”Raman spectrums” (line 270) should be spectra, we should aim for excellence in the presented work both in terms of presentation and scientific soundness.
2) The section on materials and methods is poorly prepared. No information about the source of any of the presented chemicals, materials, and purity. The authors should spend some time on this section, as it cannot and will not be accepted in the current state. All of the methods must be explained and described, and still plenty is missing.
3) The mass loading of 0.09 to 0.12 mg is extremely low, usually the mass loading for the battery devices is in the range of 10 mg per cm2, which poses the question of why not to coat the prepared slurry with copper or aluminium and check how it will behave really during electrochemical polarization. No one will use such a small material loading. Additionally, do you have any data on the electrochemical behaviour of pristine carbon paper if it would be used as the electrode itself?
4) What is missing, what is the energy and power output of such a device? Is it comparable to Li-Air ,Li-O2 batteries or other Li-CO2 batteries? (until you do not have such information/discussion). Where can it be placed in comparison to other energy storage devices?
5) I still maintain my opinion from the last review that it is necessary to understand what is happening with the cell when CO2 is introduced into the electrochemical cell. Is there any pressure increase? What are the gasses formed? Temperature changes? This experimental setup should be improved so that the introduction of CO2 can be controlled and quantified, as the moment the coin cell with a hole is not the most precise setup.
6) Is there any self-discharge issue if the cell is placed on the open circuit voltage/potential? (when there is no CO2)?
7) “Constant current charge-discharge cycles are conducted under conditions of 0.2 A·g-1 and a cut-off capacity of 1000 mAh·g-1” what about the data presented in figure 5f? Additionally, it would be beneficial to see the cyclic life graph of the devices as I understand the final cycle is the last working cycle?
As I mentioned at the beginning of the review there are still some parts of the presented work which are not corrected and despite the improvement in my opinion it is still a rejection.
Comments on the Quality of English Language
There are still some typing errors.